# Enhanced Thermal Conductivity of High-Density Polyethylene Composites with Hybrid Fillers of Flaky and Spherical Boron Nitride Particles

**DOI:** 10.3390/polym16020268

**Published:** 2024-01-18

**Authors:** Zhenliang Gao, Yiding Wang, Baogang Zhang, Li Liu, Xianhu Liu

**Affiliations:** 1Institute of High Performance Polymer, Qingdao University of Science & Technology, Qingdao 266042, China; 2National Engineering Research Center for Advanced Polymer Processing Technology, Zhengzhou University, Zhengzhou 450002, China

**Keywords:** high-density polyethylene, synergistic effect, hybrid fillers, boron nitride, thermal conductivity, electron microscopy, crystallinity

## Abstract

The synergistic effect between different fillers plays a crucial role in determining the performance of composites. In this work, spherical boron nitride (BN) and flaky BN are used as hybrid fillers to improve the thermal conductivity (TC) of high-density polyethylene (HDPE) composites. A series of HDPE composites were prepared by adjusting the mass ratio (1:0, 4:1, 2:1, 1:1, 1:2, 1:4, and 0:1) of spherical BN and flaky BN. The SEM results indicate that the spherical BN (with a particle size of 3 μm) effectively filled the gaps between the flaky BN (with a particle size of 30 μm), leading to the formation of more continuous heat conduction paths with the composite. Remarkably, when the mass ratio of spherical BN to flaky BN was set to 1:4 (with a total BN filling amount of 30 wt%), the TC of the composite could reach up to 1.648 Wm^−1^K^−1^, which is obviously higher than that of the composite containing a single filler, realizing the synergistic effect of the hybrid fillers. In addition, the synergistic effect of fillers also affects the thermal stability and crystallization behavior of composites. This work is of great significance for optimizing the application of hybrid BN fillers in the field of thermal management.

## 1. Introduction

In recent years, with the progress of science and technology, microelectronic packaging technology and integration technology have developed rapidly [1,2,3]. The output power of electronic components is increasing, but its volume is becoming smaller, resulting in a large amount of heat generated during use [4]. Heat accumulation reduces the reliability of electronic components and shortens their working life, so the requirements for the thermal conductivity (TC) of packaging materials are also increasing [5,6,7]. Polymer materials are widely used in fields such as mechanical manufacturing, electronic devices, and aerospace due to their light weight, corrosion resistance, high specific strength, easy processing, and excellent insulation performance [8,9,10,11]. However, the intrinsic TC of most polymer materials is very low (0.1~0.5 Wm^−1^K^−1^), which limits their application as thermally conductive materials [12,13,14,15]. In order to expand the application scope of polymer materials in heat conduction, it is necessary to improve their TC. At present, there are mainly two methods used to improve the TC of polymer materials [16,17,18]. The first method involves improving the crystallinity of polymer materials, increasing the orientation of molecular chain arrangements, and strengthening the interactions between molecular chains [19,20]. Although some progress has been made toward researching this method, its suitability for large-scale industrial production is limited by its harsh preparation conditions, refractory, the difficulty of forming and processing, and high cost [21]. The second method involves introducing a high-TC metal or inorganic fillers into the polymer matrix to prepare polymer-based thermally conductive composites and improve the TC of the material [22,23,24,25]. According to the thermal conductivity theory, when the filling amount is low, the fillers are not fully in contact with each other, a good heat conduction path is not formed, and heat transfer occurs mainly through the polymer matrix with a low TC. Upon further increasing the filling amount, the effective contact between the filler forms a connected heat conduction path and the heat is mainly passed through the high-TC filler. This method has the advantages of a simple production process, low preparation cost, and easy processing, so it has become the main method to improve the TC of polymer materials [26,27,28,29]. The fillers of polymer-based thermally conductive composites are mainly divided into metal nanomaterials such as silver, copper, aluminum nanoparticles, and nanowire [30,31,32,33]; ceramic nanomaterials such as BN [34,35], aluminum oxide (Al_2_O_3_) [36], silicon carbide (SiC) [37], etc.; and carbon nanomaterials such as graphite, graphene, carbon nanotubes (CNTs), and diamond [38,39,40].

To our knowledge, the influence of some thermally conductive fillers on the TC of composites has been studied. For example, Yu et al. [41] prepared composite microspheres by depositing copper particles on polystyrene microspheres. Compared with simple blends, the addition of 23 vol% copper particles had a TC of up to 26.14 Wm^−1^K^−1^ and the conductivity was also improved by eight orders of magnitude. Park et al. [42] prepared composites using long CNTs and short CNTs as fillers and EP (epoxy) as a matrix. The results show that long CNTs can improve the TC of composites more efficiently than short CNTs at the same content. When the mass fraction of a long CNT reaches 60%, the TC of the composite is as high as 55 Wm^−1^K^−1^. Among these fillers, carbon materials or metals often have high conductivity, which hinders their application in fields with insulating requirements like electronic packaging. In contrast, ceramic fillers are frequently utilized to prepare thermally conductive polymer composites due to their excellent insulation performance and high TC [43]. In ceramic fillers, BN has been widely used due to its high TC, good insulation, excellent thermal stability, and low price [44,45]. For example, Lin et al. [46] prepared composites by filling BNNs (boron nitride nanosheets) into EP. The experimental results show that when the mass fraction of BNNs is 30%, the TC of the composites increases by 316% compared to the pure EP matrix. Xie et al. [47] prepared composites by making BN distribute orientation in the PVA (polyvinyl alcohol) matrix under the action of external forces. The research shows that the higher the orientation degree of BN in the matrix, the higher the in-plane TC of the composites, and the in-plane TC of 30 wt% BN/PVA is 4.41 Wm^−1^K^−1^.

Although the introduction of BN can improve the TC of a material, a single filler requires a higher content to construct more heat conduction paths and improve the TC of the composites. According to the relevant literature, the TC of composites can be improved more efficiently by adding multi-shape thermally conductive fillers into the polymer matrix [48]. For example, Jung et al. [49] used spherical AlN (aluminum nitride) and two-dimensional BN as fillers and an EPDM (ethylene–propylene–diene monomer) rubber as matrix to prepare composites and studied the effects of differently shaped fillers on the TC of materials. The results show that when the filler volume fraction is 70% and the volume ratio of AlN to BN is 1:1, the TC of the composites is 4.76 Wm^−1^K^−1^, which is 57.10% higher than that of the AlN composites with the same volume fraction. Xiao et al. [24] prepared the composites with CNT and BN as fillers and PVDF (poly(vinylidene fluoride)) as a matrix. The results show that when the mass fraction of BN is 20% and the mass fraction of CNT is 2%, the TC of the composite increases to 1.30 Wm^−1^K^−1^, which is 34% higher than that of a single BN composite. The above results indicate that hybrid fillers are more effective in improving the TC of composites, which is mainly because the synergistic effect of hybrid fillers (commonly known as the “bridge-link effect”) promotes the formation of heat conduction paths in the polymer matrix [35]. Although the influence of hybrid fillers on the TCs of composites has been widely studied, there are few studies on the TC, thermal stability, and crystallinity behavior of composites that involve adjusting the mass ratio of fillers with different shapes. 

In this work, spherical BN (3 μm) and flaky BN (30 μm) are filled into the HDPE matrix with different mass ratios. The spherical BN may fill the gap between the flaky BN particles, forming more heat conduction paths, and a synergistic effect between the hybrid fillers on the TC of HDPE composites is expected. The fracture morphologies of the HDPE composites were observed by SEM and the crystallinity behavior was analyzed using WAXD (wide-angle X-ray diffraction) and DSC (differential scanning calorimetry). The TC of HDPE composites was measured by Hot Disk and the heat conduction rate was simulated using the finite element method. The thermal stability of HDPE composites was also studied. To our knowledge, this is the first time that the effects of mass ratios of differently shaped BN on the thermal stability, crystallization properties, and TC of composites have been studied simultaneously. This is of great significance for optimizing the application of BN fillers in the field of heat conduction.

## 2. Experimental Approach

### 2.1. Materials

HDPE 2911 with a density of 0.960 g/cm^3^ and a melt flow rate of 20.0 g/10 min was supplied by Fushun Petrochemical, Fushun, China. The maleic anhydride grafted high-density polyethylene (HDPE-g-MAH1040) with a graft ratio of 1.2 wt.% was supplied by ExxonMobil Chemical Company, Shanghai, China. HDPE-g-MAH as a compatibilizer can reduce the agglomeration of filler and contribute to the dispersion of filler in the matrix. Spherical BN (particle size of 3 μm) and flaky BN (particle size of 30 μm) were provided by Tianyuan Technology Group, Shenzhen, China. Among them, BN has the characteristics of high TC (250–300 Wm^−1^K^−1^), strong thermal stability, and excellent insulation.

### 2.2. Sample Preparation

In order to remove moisture, the matrix and filler were dried in a vacuum oven at 60 °C for 24 h before the melt blending. HDPE and HDPE-g-MAH with mass fractions of 65 wt.% and 5 wt.% were added to the internal mixer (Haake Minilab П) at 180 °C for 3 min with a rotational speed of 30 rpm. Then, the BN with a mass fraction of 30 wt.% (two different shapes of BN mixed, mass ratios of 1:0, 4:1, 2:1, 1:1, 1:2, 1:4, and 0:1) was added to the mixer and blended for 7 min under the same conditions to obtain the composites. The matrix and the solubilizer were pretreated in a high-speed dry mixer before feeding. Samples for the characterization of various properties were molded at a temperature of 180 °C and a pressure of 10 MPa for 10 min. The obtained composites were named S3/F30 X:Y, where S (F) represents spherical (flaky) BN and X:Y denotes the mass fraction ratio of 3 μm to 30 μm BN. The naming of the samples is shown in Table 1.

### 2.3. Characterization

Scanning electron microscopy (SEM, JSM-6380, JEOL, Tokyo, Japan) was used to characterize the dispersion of BN in the composites under 30 kV accelerating voltage. The fracture surfaces of all samples were coated with Au prior to characterization. A thermal gravimetric analyzer (TGA, Pyris 1, Perkinelmer, Waltham, MA, USA) was used from 30 to 900 °C in a N_2_ atmosphere at a heating rate of 10 °C/min to compare the thermal stability of all samples. Differential scanning calorimetry (DSC, Q2000, TA, New Castle, DE, USA) was performed to investigate the melting and crystallization behaviors of all samples. Samples of about 5 mg were heated from 25 °C to 190 °C at a rate of 10 °C/min under a N_2_ atmosphere and the crystallinity ($X_{c}$) of the samples was calculated using Equation (1):(1)$$X_{c}=\frac{\;{\Delta H}_{m}}{{\Phi \Delta H}_{m}^{0}}$$
where ${\Delta H}_{m}$ is the melting enthalpy of the sample and ${\Delta H}_{m}^{0}$ is the melting enthalpy of HDPE 100% crystallization with a value of 289.9 J/g. The Φ is the relative mass fraction of HDPE in the composite. An X-ray setup (Bruker D8 Discover, Bruker, Saarbrucken, Germany) equipped with a Cu Kα (λ = 1.54 Å) X-ray source was used to record the WAXD measurements at room temperature. The TC of all samples was measured with a Transient Hot Disk TPS 2500S instrument (Hot Disk AB, Gothenburg, Sweden). Before measurement, the probe and sample surface were carefully cleaned with ethanol. 

## 3. Results and Discussion

Figure 1 shows the fracture morphologies obtained using SEM after the brittle fracture of hybrid BN-filled HDPE composites, and the total filling amount of BN was 30 wt%. According to previous studies, the formation of the heat conduction path is related to the overlap between the fillers. That is to say, the better the overlap between the thermally conductive fillers, the better the thermal conductivity performance. Figure 1a,g show the fracture morphologies of S3/F30 1:0 and S3/F30 0:1, respectively. It can be seen that the latter shows much better overlapping of fillers than the former, which means that the TC is also better. Figure 1b,c show the fracture morphologies of samples S3/F30 4:1 and S3/F30 2:1, respectively. In this case, the content of BN with a small particle size is greater than that with a large particle size. It can be seen that the overlap between fillers is not ideal, and not many heat conduction paths are formed. The reason is that small-particle-size BN is not conducive to the formation of heat conduction paths, and large-particle-size BN is not conducive to filling the gap between small particle size BNs. Figure 1d shows the fracture morphology of sample S3/F30 1:1. At this time, the content of small-particle-size BN is the same as that of large-particle-size BN, and it can be seen that the heat conduction paths of the composite are obviously improved. Figure 1e,f show the fracture morphologies of samples S3/F30 1:2 and S3/F30 1:4, respectively. It can be seen that the overlap between fillers is good and more heat conduction paths are formed, especially for sample S3/F30 1:4. This is because the large-size BN shortens the distance between the fillers, and the small-size BN more easily fills the gap between the large-size BN, thus forming more heat conduction paths. Figure 1h,i show the micromorphologies of spherical BN and flaky BN, respectively. In addition, the BN in all composites is uniformly distributed in the HDPE matrix, and there is no large area of aggregated filler or a large number of pores left due to BN shedding, which is attributed to the effect of solubilizer HDPE g-MAH [35]. According to our previous work, the addition of HDPE-g-MAH plays a key role in reducing filler aggregation, which helps to build up the thermal conduction paths in the matrix.

In order to study the influence of hybrid BN on the thermal stability of composites, the thermal stability of all samples was characterized by TGA. The results are shown in Figure 2 and Table 2. T_5%_ (the temperature at which 5% weight loss occurs) and T_MAX_ (the peak of the DTA curve) were used to analyze the TGA curves. As can be seen from Figure 2a, the composites with a single BN filler exhibit one-step decomposition behavior in a nitrogen atmosphere, which corresponds to the decomposition of HDPE and HDPE-g-MAH (400–550 °C). With a change in mass ratio of hybrid BN, the composites still present one-step decomposition behavior, which indicates that the mass ratio of the two types of BN does not affect the thermal decomposition behavior of the HDPE matrix. After complete pyrolysis, HDPE becomes a gas single-phase, while BN (even when mixed with two types of BN) does not decompose in a nitrogen atmosphere at 900 °C. Therefore, the remaining material at the end of a TG experiment is BN. It can be seen that the mass fraction of the remaining material is around 30%, which is consistent with the actual BN mass fraction filled. It can be seen from Table 2 that the T_5%_ value of sample S3/F30 1:0 is the largest at 446.14 °C. With the introduction of flaky BN, the T_5%_ value of the composite decreases slightly, but the T_5%_ value of sample S3/F30 1:4 is consistent with that of S3/F30 1:0. This may be due to the dense network structure formed by the hybrid BN in the composite S3/F30 1:4, which effectively prevents the evaporation and diffusion of matrix degradation products. In addition, the T_MAX_ of composites with hybrid BN is higher than that of composites with single BN. For example, the T_MAX_ of sample S3/F30 1:4 is 6.07 °C higher than that of S3/F30 1:0 and 10.85 °C higher than that of S3/F30 0:1.

HDPE is a commonly used semi-crystalline polymer. In order to study the crystalline structure of HDPE and HDPE/BN composites, WAXD tests were conducted. Figure 3a displays the 1D-WAXD curves of HDPE composites with different mass ratios of hybrid BN. The semi-crystalline phase of HDPE and the single crystalline phase of BN can be seen, where the two characteristic peaks of 2θ = 21.37° and 2θ = 23.51° represent diffraction in the HDPE (110) and (200) planes, and the characteristic peaks of 2θ = 26.13° represent diffraction in the BN (002) plane. When the mass ratio of hybrid BN changes, no other diffraction peaks appear, indicating that the mass ratio of hybrid BN has no effect on the crystal form.

The melting and crystallization behaviors of the composites were characterized by differential scanning calorimetry (DSC). The heating and cooling curves of all samples are displayed in Figure 3b,c. It can be seen from the results in Figure 3b and Table 2 that the melting temperature (T_M_) of all the composites ranges from 130.97 to 132.77 °C, which is lower than that of pure HDPE (134.73 °C). However, the T_M_s of all the composites are similar, indicating that the mass ratio of hybrid BN has little effect on the T_M_ of the composites. From the results in Figure 3c and Table 2, it can be found that the crystallization temperature (T_C_) of all the composites ranges from 122.14 to 124.16 °C, which is higher than that of pure HDPE (117.79 °C). However, the T_C_s of all the composites are similar, indicating that the mass ratio of hybrid BN has little influence. In addition, the crystallinity (X_C_) of hybrid BN composites is higher than that of single BN composites. For example, the crystallinity calculated by DSC shows that the X_C_ of sample S3/F30 1:4 is 74.1%, while the X_C_s of samples S3/F30 1:0 and S3/F30 0:1 are 71.1% and 71.4%, respectively. The results of the WAXD calculations have the same rule. This is mainly because BN has heterogeneous nucleation and hybrid BN provides more nucleation points for the crystallization of HDPE [50], which is beneficial for improving the X_C_ of the composites. Generally speaking, the TC of polymers is directly proportional to their crystallinity, as the chain arrangement in the crystallization region is more regular and can form more heat conduction paths. However, in this study, the change in crystallinity is not the only factor that affects the change in TC, so it is necessary to explore other factors that lead to changes in the TC of composites.

The TC of the composites in the direction of thickness was measured by the Hot Disk, as shown in Figure 4. Figure 4a shows the relationship between the TC of the composites and the mass ratio of the hybrid BN. It can be seen that the TC of sample S3/F30 1:0 is the lowest, and with the addition of flaky BN, the TC of the composites first increases and then decreases with the increase in flaky BN mass ratio. For example, when the mass ratio of S3/F30 changes from 1:0 to 1:4, the TCs of the composites increase from 0.736 Wm^−1^K^−1^ to 1.648 Wm^−1^K^−1^, which is 282.27% higher than that of pure HDPE (0.431 Wm^−1^K^−1^). However, when all the fillers in the composite are flaky BN, the TC of the composite decreases to 1.479 Wm^−1^K^−1^. The reason for this phenomenon is that when there are more large-particle-size flaky BNs in the composite, which shortens the distance between the fillers in the matrix, and a few small-particle-size spherical BNs can fill the gap between the large-particle-size fillers, forming more heat conduction paths, the average free path of phonons is increased and heat loss in the heat flow process is reduced [44]. Therefore, the TC of the composites has been greatly improved. In addition, in order to compare the TC of composites with that of pure HDPE, the thermal enhancement factor (φ) was introduced in this study. It was defined as follows:(2)$$\varphi =\frac{\lambda_{1}-\lambda_{0}}{\lambda_{0}}\times 100\%\;$$
where $\lambda_{1}$ and $\lambda_{0}$ represent the TCs of the composites and pure HDPE, respectively. The variation in the thermal enhancement factor (φ) is shown in Figure 4b. It can be seen that φ increases first and then decreases with increases in the mass ratio of flaky BN.

In order to verify the above results, the finite element method was used to simulate the influence of the hybrid BN mass ratio on the heat conduction rate of the composite [44]. Here, a three-dimensional model of HDPE/BN composite with the size of 80 × 80 × 80 μm was established. We have listed the three-dimensional models of samples S3/F30 1:4 and S3/F30 4:1, as shown in Figure 5a,b. From the model, it can be seen that spherical BN and flaky BN are randomly distributed in the HDPE matrix, similar to solid blocks. Firstly, the content of hybrid BN and the size of the matrix were determined, and then, the influence of BN with different mass ratios on the thermal conductivity was simulated by changing the parameters of injected particles. In this work, the total BN content of the composite is 30 wt%, and we set the mass ratio of spherical BN to flaky BN to 4:1, 2:1, 1:1, 1:2, and 1:4. Therefore, the corresponding mass fractions of flaky BN are 6 wt%, 10 wt%, 15 wt%, 20 wt%, and 24 wt%, respectively. For all models, a heating table with a constant temperature was set on the bottom surface, and five time nodes (t = 0 ms, t = 0.6 ms, t = 1.2 ms, t = 1.8 ms, and t = 2.4 ms) were selected to observe the temperature distribution of the whole model. It can be seen from Figure 5c that the heat conduction rate of the composite increases when the mass fraction of flaky BN increases within the same time interval. Among them, the upper surface temperature of S3/F30 1:4 is higher than that of any other model, indicating that when the mass ratio of S3 to F30 is 1:4, the thermal conductivity of the composite is the best, which is consistent with the measured results.

## 4. Conclusions

In this work, a series of HDPE/BN composites were prepared by melt blending. The effects of the mass ratio of hybrid BN on the microstructure, thermal stability, crystallization properties, and TC of the composites were systematically investigated. The dispersion and microstructure of fillers in the composite were studied through SEM. The results show that BN particles were uniformly dispersed in the HDPE matrix, and there was no large area of filler aggregation or a large number of pores. Studies on the crystal structure of the matrix show that the hybrid BN contributes to the improvement of the crystallinity of the composite because the hybrid BN provides more nucleation points for the crystallization of HDPE. Thermogravimetric analysis shows that hybrid BN is beneficial for the improvement of the thermal stability of the composites. In addition, the hybrid BN is beneficial for the improvement of the TC of the composite, and the maximum increment of the thermal conductivity relative to the matrix (1.648 Wm^−1^K^−1^) is 282.37%. This is because a few small-particle-size spherical BN can fill the gaps between the large-particle-size fillers, forming more heat conduction paths, thus increasing the average free path of phonons and reducing heat loss in the heat flow process. The results of the finite element conduction simulation are also consistent with the experimental results.

## Figures and Tables

**Figure 1 polymers-16-00268-f001:**
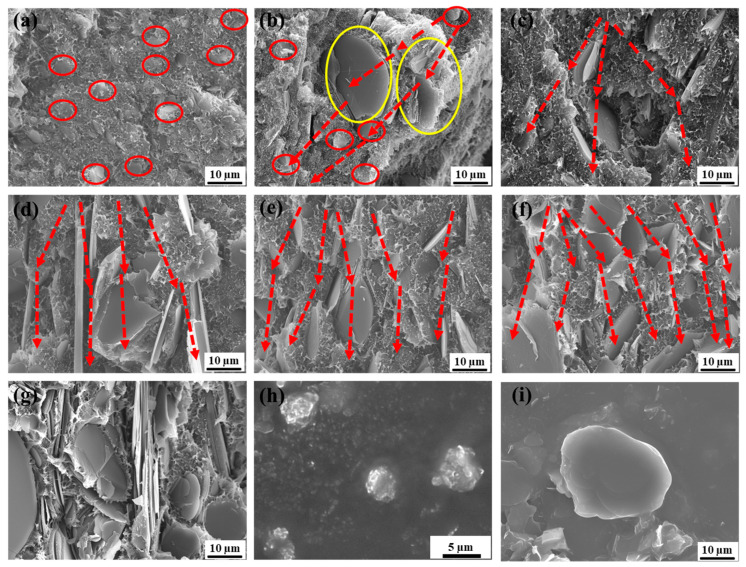
SEM images of the fracture morphologies of composites with 30 wt% BN: (**a**) S3/F30 1:0, (**b**) S3/F30 4:1, (**c**) S3/F30 2:1, (**d**) S3/F30 1:1, (**e**) S3/F30 1:2, (**f**) S3/F30 1:4, (**g**) S3/F30 0:1. SEM images of BN particles: (**h**) 3 µm, (**i**) 30 µm. In the image, the red circle represents the spherical BN, the yellow circle represents the flaky BN, and the red arrow represents the heat conduction path.

**Figure 2 polymers-16-00268-f002:**
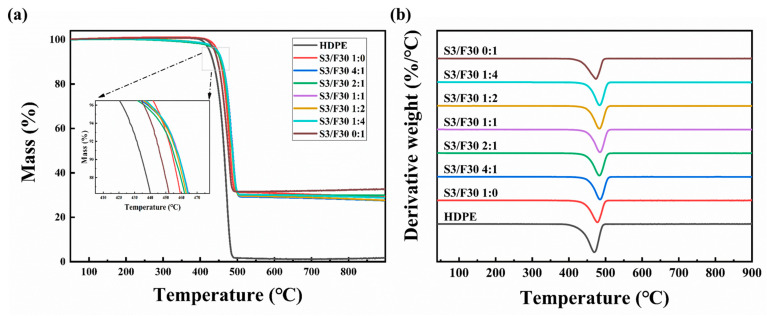
(**a**) TGA and (**b**) differential thermal analysis (DTA) curves of HDPE composites.

**Figure 3 polymers-16-00268-f003:**
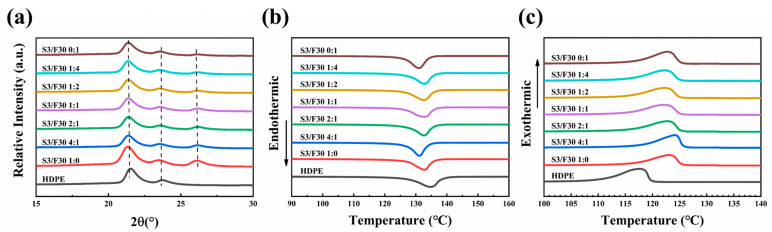
WAXD and DSC results of HDPE composites: (**a**) WAXD profiles, (**b**) heating curve, (**c**) cooling curve.

**Figure 4 polymers-16-00268-f004:**
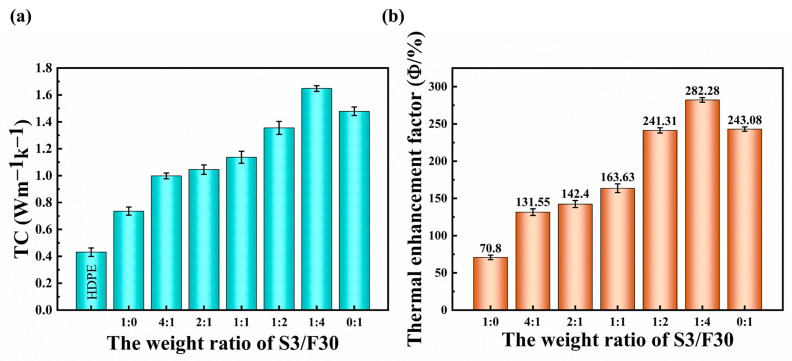
The changes in (**a**) TC and (**b**) thermal enhancement factor of HDPE composites.

**Figure 5 polymers-16-00268-f005:**
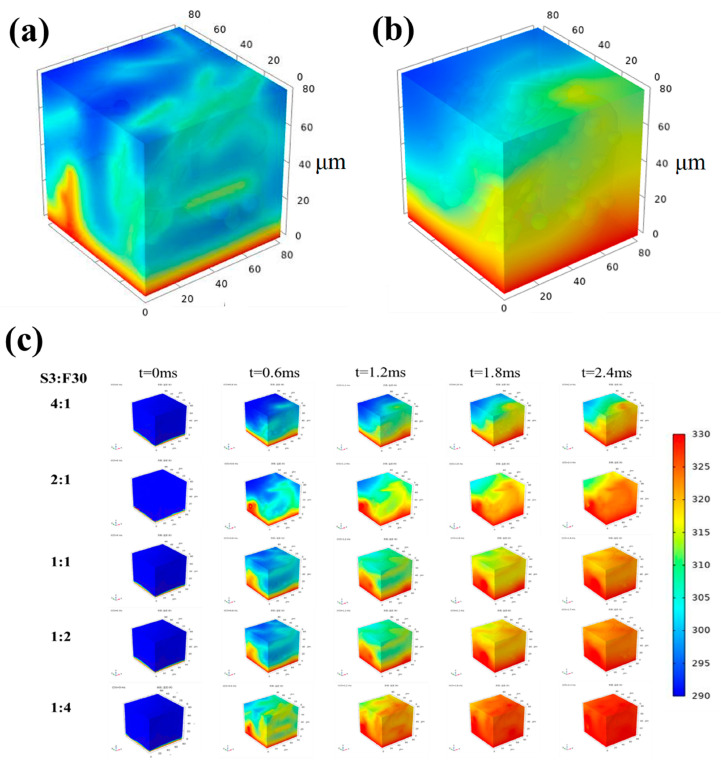
The three-dimensional models of (**a**) S3/F30 1:4 and (**b**) S3/F30 4:1, and (**c**) the simulation of temperature variation in composites over the same time interval.

**Table 1 polymers-16-00268-t001:** The naming of samples.

Mass Ratio of S3/F30	1:0	4:1	2:1	1:1	1:2	1:4	0:1
Sample	S3/F30 1:0	S3/F30 4:1	S3/F30 2:1	S3/F30 1:1	S3/F30 1:2	S3/F30 1:4	S3/F30 0:1

**Table 2 polymers-16-00268-t002:** Thermal properties and melting and crystallization parameters of HDPE composites.

Sample	HDPE	S3/F30 1:0	S3/F30 4:1	S3/F30 2:1	S3/F30 1:1	S3/F30 1:2	S3/F30 1:4	S3/F30 0:1
T_5%_/°C	424.79	446.14	443.58	441.09	443.58	444.04	446.10	438.58
T_MAX_/°C	467.35	479.85	484.68	482.72	484.71	485.03	485.92	475.07
T_M_/°C	134.73	132.77	131.16	132.62	132.70	132.53	132.25	130.97
T_C_/°C	117.79	123.02	124.16	122.69	122.14	122.37	122.25	122.76
X_C_/% (WAXD)	66.9	68.9	73.6	72.7	71.3	72.4	73.9	69.5
X_C_/% (DSC)	67.3	71.1	74.8	73.6	72.5	71.8	74.1	71.4

## Data Availability

Data are contained within the article.

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
