# Peer review of "Enhanced Thermal Conductivity of High-Density Polyethylene Composites with Hybrid Fillers of Flaky and Spherical Boron Nitride Particles"

_polymers, 2024, doi:10.3390/polym16020268_

Round 1

Reviewer 1 Report

Comments and Suggestions for Authors

This article is dedicated to an important and relevant topic. However, the scientific novelty of the results obtained is questionable and should be stated in the Introduction. A description of the influence of the shape and distribution of BNs particles, as well as the combination of fillers with different form factor can be extended in the Introduction.

In addition, there are a lot of technical questions that need answers or corrections:

1. Why did you used filler (a mixture of spherical BN and flaky BN) mass fraction of 30 wt.%?

2. Figure 1. – There are no spherical particles in the figure, or they are difficult to identify. Addition of SEM images of initial fillers can improve the paper. It also should be adding the SEM of S3/F30 1:0 and S3/F30 0:1.

3. In lines 146, 147: “It can be seen that the overlap between fillers is not ideal, and there are not many heat conduction paths formed.” – it should be described in more detail or highlighted in the figure. As well as further down the text, when describing "heat conduction pathways".

4. Figure 2 and Table 2 – is the result of TGA/DTA very important? The duplication in the figure and table seems unnecessary, especially as the authors write: “the mass ratio of the two types of BN does not affect the thermal decomposition behavior of HDPE matrix“.

5. Figure 2 and Figure 3 will be more informative if pure polymer and initial fillers are added.

6. Lines 212-214: “… the TC of polymers is directly proportional to their crystallinity, as the chain arrangement in the crystallization region is more regular and can form more heat conduction paths” – it is desirable to add references to the literature where this information is given. I suppose it is not always "directly", and not always through "heat conduction paths".

7. line 221: “It can be seen that the TC of sample S3/F30 is the lowest” – Which one? The wording needs to be clarified.

8. Figure 4(a). – TC of pure HDPE should be added.

9. Description of results, obtained with the finite element method is very poor.

- Lines 243, 244 – “From the model, it can be seen that spherical BN and flaky BN are randomly distributed in the HDPE matrix, similar to solid blocks” – where can we see this result? In Fig. 5 there is no particles distribution.

- In the Fig. 5 indicated t=0ms, t=0,6ms... is it a time of heating?

 - Line 247 "... the mass fraction of sheet BN is 6 wt%, 10 wt%, 15 wt%, 20 wt% and 24 wt%" - is this the concentration of a mixture of two fillers (spherical BN and flaky BN) or what is it? Where are these values in Figure 5?

Description of Fig. 5 should be strongly improved.

10. It is necessary to add a comparison of the results obtained with the world level. Especially to compare the achieved value of TC (1.648 Wm-1K-1) with the results presented in other scientific articles.

In general, there are many papers describing percolation effects in thermally conductive polymers. Taking into account the high rating of the “Polymers” I can recommend improving the paper by at least some description of the thermal conductivity model or calculation of the percolation threshold instead of the term "heat conduction paths".

Reviewer 2 Report

Comments and Suggestions for Authors

The manuscript can be improved. Some comments: 

1. Include the expansion of abbreviations when they are used for the first time. For example, SEM, EP, BNN, PVA, EPDM, PVDF, WAXD, DSC etc. were used without their expansion. 

2. highly conductivity, not a good usage. 

3. Inherent properties of both BN should be provided. Most importantly, the thermal conductivity of the fillers. 

4. Figure 1: How did you understand the difference between S3 and F30 from those images? For me, except (a) all other images seem to be identical. If possible mark S3 and F30 in the image. 

5. line 221: TC of sample S3/F30 is lowest... Something missing here. 

6. systematic investigated?

7. Figure 4. (a) and (b) are the same. Only one is necessary. Include the TC of HDPE alone in (a) then no need for (b). Also, do you think the change from 0.431 to 1.648 is great? Is it appropriate to convert as equivalent to 282% of change? Does this change have any practical significance and application?

Comments on the Quality of English Language

Minore correction to English phrases needed. 

Author Response

The manuscript can be improved. Some comments:

  1. Include the expansion of abbreviations when they are used for the first time. For example, SEM, EP, BNN, PVA, EPDM, PVDF, WAXD, DSC etc. were used without their expansion. 

Reply: Thank you very much for your comments, which have been corrected and highlighted in red in the original article.

  1. highly conductivity, not a good usage. 

Reply: Thank you very much for your suggestion. The original article has been changed to " Among these fillers, carbon materials or metals often have high conductivity, which hinders their application in fields with insulating requirements like electronic packag-ing " and marked in red font.

  1. Inherent properties of both BN should be provided. Most importantly, the thermal conductivity of the fillers.

Reply: Thanks for your suggestion, the inherent properties of BN have been added to the original article and marked with red font.

  1. Figure 1: How did you understand the difference between S3 and F30 from those images? For me, except (a) all other images seem to be identical. If possible mark S3 and F30 in the image. 

Reply: We are very sorry to cause you such misunderstanding. We have made modifications to the original image and marked it accordingly.

  1. line 221: TC of sample S3/F30 is lowest... Something missing here. 

Reply: Thank you very much for your comments, which have been corrected and highlighted in red in the original article.

  1. systematic investigated?

Reply: Thank you very much for your valuable advice. We have conducted a systematic study on polymer-based thermal conductive composites in the introduction, and the research indicate that hybrid fillers are more effective in improving the TC of composites, which mainly because the synergistic effect of hybrid fillers (commonly known as “bridge-link effect”) promotes the formation of the heat conduction paths in the polymer matrix.

  1. Figure 4. (a) and (b) are the same. Only one is necessary. Include the TC of HDPE alone in (a) then no need for (b). Also, do you think the change from 0.431 to 1.648 is great? Is it appropriate to convert as equivalent to 282% of change? Does this change have any practical significance and application?

Reply: Thank you very much for your suggestion. First of all, Figure 4. (a) shows the TC of all samples, while the the thermal enhancement factor (φ) introduced in Figure 4. (b) can intuitively see the TC change of the composites relative to pure HDPE. Second, the internal phonon scattering of polymer materials is very serious, so it is not easy to improve its TC. As can be seen from the introduction, the TC of the composite prepared in this paper has been greatly improved, which will broaden its application in the field of thermal management.

Reviewer 3 Report

Comments and Suggestions for Authors

In general, the topic addressed in the paper Enhanced thermal conductivity of HDPE composites with hybrid fillers of flaky and spherical boron nitride particles” by Z. Gao, et al. is very interesting. The paper shows novel results in the field of Thernal conductivity in HDPE . Meanwhile the paper needs minor changes in order to be suitable for publication in Polymers.

Please find below a list of observations:

1.Abstract: it is important for readers to know the concentrations of boron nitride, only the 1:4 ratio is mentioned. The particle size should also be mentioned, since voids are filled in the composites.

2. Keywords: add the terms "High density polyethylene", "Electron microscopy" and "Crystallinity".

3. Figure 1: it is recommended to use geometric shapes to show the surface features of the composite materials, e.g., circles, arrows, lines, etc. For example, line 146 states "it can be seen that the overlap between fillers...", which can be indicated by arrows in the SEM image.

4. Figure 4: add the error bars in Figure 4b.

Author Response

In general, the topic addressed in the paper “Enhanced thermal conductivity of HDPE composites with hybrid fillers of flaky and spherical boron nitride particles” by Z. Gao, et al. is very interesting. The paper shows novel results in the field of Thernal conductivity in HDPE. Meanwhile the paper needs minor changes in order to be suitable for publication in Polymers.

Please find below a list of observations:

  1. Abstract: it is important for readers to know the concentrations of boron nitride, only the 1:4 ratio is mentioned. The particle size should also be mentioned, since voids are filled in the composites.

Reply: Thank you very much for your valuable advice, which have been corrected and highlighted in red in the original article.

  1. Keywords: add the terms "High density polyethylene", "Electron microscopy" and "Crystallinity".

Reply: Thank you very much for your suggestion. We have revised and marked in red in the original article.

  1. Figure 1: it is recommended to use geometric shapes to show the surface features of the composite materials, e.g., circles, arrows, lines, etc. For example, line 146 states "it can be seen that the overlap between fillers...", which can be indicated by arrows in the SEM image.

Reply: Thank you very much for your valuable advice. We have optimized as required, as shown in Figure R1.

  1. Figure 4: add the error bars in Figure 4b.

Reply: Thank you very much for your suggestion. We have added the error bar to Figure 4b.

Reviewer 4 Report

Comments and Suggestions for Authors

In this manuscript, the authors studied the effect of the spherical boron nitride (BN) and flaky BN as hybrid fillers into high density polyethylene (HDPE). The results of SEM indicate that spherical BN can fill the gaps between flaky BN, forming more continuous heat conduction paths. When the mass ratio of spherical BN to flaky BN is 1:4 (with a total BN filling amount of 30 wt%), the thermal conductivity of the composite can reach 1.648 Wm-1K-1. In addition, the synergistic effect of fillers also affects the thermal stability and crystallization behavior of composites. The interpretations of the results were deeply discussed. The quantity and quality of the figures are appropriate. We believe that this research can be utilized for optimizing the application of hybrid fillers in the field of thermal management. Summary: I recommend this manuscript after considering my comments as; Line 64, what does BNNs mean? Line 76, what does AIN mean? Line 78, what does PVDF mean? Line 130, if it is possible, the authors can determine the degree of crystallinity from WAXD and compare them with those from DSC. Line 131, AHmo was not defined? Line 214-216 “However, in this study, the change law of TC is not consistent with that of crystallinity, so it is necessary to explore the factors that lead to significant changes in the TC of composites.” This sentence is not clear where the change of the crystallinity in your samples from 71.1 to 74.8 %, i.e. the change is about 5% while the change on the TC is about 124%. Line 293, the page number was missed. Line 295, the page number was missed. Line 300, the page number was missed. Line 304, the page number was missed. Line 323, the page number was missed. Line 325, the page number was missed. Line 329, the page number was missed. Line 334, the page number was missed. Line 346, the page number was missed. Line 351, the page number was missed. Line 361, the page number was missed. Line 363, the page number was missed. Line 383, the page number was missed. Line 385, the page number was missed. Line 395, the page number was missed.
